# The Neurovascular Unit Dysfunction in Alzheimer’s Disease

**DOI:** 10.3390/ijms22042022

**Published:** 2021-02-18

**Authors:** Luis O. Soto-Rojas, Mar Pacheco-Herrero, Paola A. Martínez-Gómez, B. Berenice Campa-Córdoba, Ricardo Apátiga-Pérez, Marcos M. Villegas-Rojas, Charles R. Harrington, Fidel de la Cruz, Linda Garcés-Ramírez, José Luna-Muñoz

**Affiliations:** 1Facultad de Estudios Superiores Iztacala, UNAM, Mexico City 54090, Mexico; alemarttlu@gmail.com; 2Neuroscience Research Laboratory, Faculty of Health Sciences, Pontificia Universidad Catolica Madre y Maestra, Santiago de los Caballeros 51000, Dominican Republic; 3Departamento de Fisiología, Escuela Nacional de Ciencias Biológicas, Instituto Politécnico Nacional, Mexico City 07738, Mexico; berecordoba21@gmail.com (B.B.C.-C.); rapatigap@gmail.com (R.A.-P.); flacruz90@hotmail.com (F.d.l.C.); adnil_gr@yahoo.com.mx (L.G.-R.); 4National Dementia BioBank, Ciencias Biológicas, Facultad de Estudios Superiores, Cuautitlán, UNAM, Mexico City 53150, Mexico; 5Unidad Profesional Interdisciplinaria de Biotecnología del Instituto Politécnico Nacional (UPIBI-IPN), Mexico City 07340, Mexico; marcusvillegas3@gmail.com; 6School of Medicine, Medical Sciences and Nutrition, University of Aberdeen, Aberdeen AB25 2ZD, UK; c.harrington@abdn.ac.uk; 7Banco Nacional de Cerebros-UNPHU, Universidad Nacional Pedro Henríquez Ureña, Santo Domingo 2796, Dominican Republic

**Keywords:** blood-brain barrier, astrocytes, microglia, amyloid peptide, Alzheimer’s disease, tau protein

## Abstract

Alzheimer’s disease (AD) is the most common neurodegenerative disease worldwide. Histopathologically, AD presents with two hallmarks: neurofibrillary tangles (NFTs), and aggregates of amyloid β peptide (Aβ) both in the brain parenchyma as neuritic plaques, and around blood vessels as cerebral amyloid angiopathy (CAA). According to the vascular hypothesis of AD, vascular risk factors can result in dysregulation of the neurovascular unit (NVU) and hypoxia. Hypoxia may reduce Aβ clearance from the brain and increase its production, leading to both parenchymal and vascular accumulation of Aβ. An increase in Aβ amplifies neuronal dysfunction, NFT formation, and accelerates neurodegeneration, resulting in dementia. In recent decades, therapeutic approaches have attempted to decrease the levels of abnormal Aβ or tau levels in the AD brain. However, several of these approaches have either been associated with an inappropriate immune response triggering inflammation, or have failed to improve cognition. Here, we review the pathogenesis and potential therapeutic targets associated with dysfunction of the NVU in AD.

## 1. Introduction

The neurovascular unit (NVU) is defined as a complex functional and anatomical structure composed of: (a) neurons and interneurons, (b) glial cells such as microglia, astrocytes, and oligodendrocytes, (c) vascular cells such as endothelial cells, pericytes, and smooth muscle cells (SMCs), and (d) a basal lamina formed by brain endothelial cells and extracellular matrix (Figure 1) [1,2]. The NVU components are intimately linked to each other, enabling an efficient system for cerebral blood flow (CBF), maintenance of neuronal metabolic activity [3], and an effective blood-brain barrier (BBB). The BBB is a specialized structure in the cerebral vasculature formed by astrocytes, pericytes, and specialized junctions such as tight-junctions (TJs) and adherent-junctions (AJs) of endothelial cells [4]. The BBB serves to limit the entry of pathogens, toxic agents, and blood cells into the brain parenchyma [5]. The junctional components of the NVU are made by gap junctions and adhesion molecules such as integrins and cadherins [6]. This interplay facilitates the influx/efflux of Ca^2+^, K^+^, and the neuromodulatory action of ATP [7]. Each NVU component plays an active and specific role in maintaining the dynamic linkages reciprocally under physiological conditions [8]. Dysregulation of the NVU and BBB are critical pathophysiological events in neurodegenerative diseases, including Alzheimer’s disease (AD) [9].

AD is one of the most important neurodegenerative disorders worldwide and the leading cause of cognitive and functional decline in the elderly [10]. The classic neuropathological hallmarks of AD involve aggregates of tau and amyloid β-peptides (Aβ). Neurofibrillary tangles (NFTs) and hyperphosphorylated tau accumulate intracellularly and are typically accompanied by neuronal loss [11,12]. The second hallmark is the presence of amyloid β-peptide (Aβ) deposits in the brain parenchyma and around cerebral blood vessels as neuritic plaques (NPs) and cerebral amyloid angiopathy (CAA), respectively. CAA is characterized by Aβ deposition within the walls of cortical and leptomeningeal arteries and veins [13], which lead to NVU dysfunction [11,12]. CAA is universally found in AD brains [14] and has been linked to neuroinflammation, chronic hypoperfusion, ischemia, and loss of the blood vessel wall integrity and hemorrhage [15,16].

In this study, we review the relationship between the NVU and BBB dysregulation and AD pathology. Furthermore, possible therapeutic targets aimed at preventing or restoring NVU in AD are discussed.

## 2. Cellular and Structural Components of the NVU Along the Cerebrovascular Tree

An extensive system of arteries, arterioles, and capillaries, delivers oxygenated blood and nutrients to the brain. The pial arteries derived from arteries go along the brain surface and penetrate the parenchyma, branching into arterioles and capillaries [17]. The cellular NVU composition changes along the cerebrovascular tree (Figure 1a):(a)Pial arteries consist of multiple layers of SMCs, separated from the endothelium by a notable elastic lamina (Figure 1b), and innervated by nerve fibers formed from sensory and peripheral autonomic ganglia [18]. The pial arteries penetrating the brain are surrounded by the subarachnoid space (SAS; Figure 1b). Traditionally, the pial arteries have been associated with CBF and neuronal homeostasis [19].(b)Penetrating arterioles have several thin SMC layers that become a single layer (Figure 1c) [20]. The density of perivascular nerves is low at this level, and the elastic lamina becomes less prominent [20]. The perivascular space (also known as “Virchow-Robin space”) is delimited by the astrocytic end-foot (glia limitans) and the vascular basement membrane [21]. It comprises different cell types, including perivascular macrophages (PVMs), pial cells, Mato cells and mast cells, and collagen and nerve fibers [21] (Figure 1c). The perivascular space is an exchange pathway for the glymphatic system. The glymphatic system has been defined as a network of perivascular pathways that favors the exchange of both solutes and liquids between the cerebrospinal fluid (CSF) and the interstitial compartments, promoting clearance of metabolites, proteins, and debris from the brain interstitium [22,23,24]. This clearance depends mainly on the aquaporin-4 water channels (AQP4), found in astrocytic end-feet [25]. It has been demonstrated that the pial and penetrating arterioles can regulate arterial tone by the extrinsic and intrinsic innervations, respectively [26].(c)The intraparenchymal arterioles are formed when the arterioles invade deeper into the brain. Here, the glial limitans and the vascular basement membrane fuse, eliminating the Virchow-Robin space [21]. These arterioles have a single layer of SMCs, lack perivascular nerves, and are encapsulated by the astrocytic end-feet (Figure 1d) [20]. Endothelial cells extend their protrusions to SMCs and connect through gap junctions [27]. They are related to functional hyperemia, which ensures a rapid increase in the CBF rate to activated brain structures [28].(d)Capillaries are the smallest vessels in the brain and exchange molecules between blood and brain across the BBB [29]. Pericytes replace the SMCs, and mural cells are immersed in the endothelial basement membrane (Figure 1e) [30]. The border of the capillaries is enveloped by astrocytic end-feet, and the neural processes can be adjacent to the capillary basal lamina (Figure 1e) [31]. Pericytes and endothelial cells make direct interdigitated contacts where cytoplasmic protrusions (pegs) of one cell type insert into the opposing cell membrane (socket) of the other cell type [32].

## 3. The Physiological Characteristics of the Blood-Brain Barrier (BBB)

The endothelial cells that make up the brain capillaries are tightly sealed and act as a barrier, referred to as the “blood-brain barrier” (BBB). The BBB is a specialized brain endothelial composition of the neurovascular system that is completely differentiated. Together with astrocytic end-feet, pericytes, and microglia, the BBB separates the circulating blood components from neurons [2,33] (Figure 1e and Figure 2a).

The nearness of the different cell types with one another in the NVU allows for an effective paracrine regulation, the maintenance of the BBB, and the normal functioning of the central nervous system (CNS), such as synaptic transmission and remodeling, neurogenesis, and angiogenesis in the adult brain [2]. The cell-to-cell hermetic contact confers to the BBB the properties of low paracellular and transcellular permeability and high transendothelial electrical resistance [29].

The firmly sealed endothelium limits the entry of most blood-derived particles into the brain unless they possess receptors or carriers (glucose, hormones, amino acids, and nucleotides) to be transported across the BBB [34]. The concentration gradient is the most considerable factor for carrier-mediated transport across the BBB and is preferentially influenced by the affinity, size, and physiochemical properties of each specific molecule [35]. The BBB excludes typically free exchange of solutes between blood-brain and brain-blood [36]. One exception is that of small lipid-soluble molecules <400 Da, which can cross the BBB via lipid-mediated diffusion [37], adsorptive endocytosis, and receptor-mediated endocytosis [38]. On the other hand, transendothelial passive diffusion enables the small influx of lipophilic and nonpolar molecules into brains across the lipid bilayer of endothelial cells [38].

Endothelial cells are typically connected at a junctional complex by the tight-junctions (TJs) and adherent-junctions (AJs) (Figure 2a) [39]. The TJs provide low paracellular permeability, high electrical resistance [40], and binding to the cytoskeleton. Claudin (Figure 2a), a transmembrane protein of 207–305 amino acids, is the main structural component of the TJs [41]. Different claudin isoforms are expressed in brain endothelial cells, such as claudin-1, -3, -5, and -12; claudin-5 being the most highly expressed in these cells [12]. Occludin (Figure 2a), another tetraspan transmembrane protein of 522 amino acids of TJs, regulates BBB integrity and permeability [42]. The transmembrane TJ proteins are linked to the actin cytoskeleton through the zonula occludens-1 (ZO-1; Figure 2a), a membrane-associated protein [43]. Together, these complex of TJ proteins decrease the paracellular diffusion and limit transcellular activity [12].

The AJs are typically found intermingled with the TJs and contribute to regulating the BBB permeability and leukocyte extravasation [44]. Vascular endothelial (VE)-cadherin is an endothelial-specific integral membrane protein, connected to the cytoskeleton via catenins (Figure 2a). The localization and expression of β-catenin, α-catenin, and p120cas, are essential for the functionality of AJs [40]. Neutrophils and lymphocytes can ingress from the blood into the brain regulated by BBB and low levels of leukocyte adhesion molecules (LAMs) [45]. Therefore, this property may prevent access of immune cells from blood to the brain, affording immunologic benefit in the CNS [45].

BBB integrity is essential for controlling the molecular composition of brain interstitial fluid (ISF), which is indispensable for correct information processing, synaptic operating, functioning, and neuronal connectivity [34]. BBB breakdown could trigger an increase in vascular permeability, reduce CBF and impaired hemodynamic responses [17]. Therefore, BBB dysfunction could facilitate the entry of toxic blood-derived molecules and cells and trigger an immune response associated with neuroinflammation, which ends in degeneration of the NVU (Figure 2b and Figure 3 steps 2c, 3 and 4).

## 4. Dysfunction of the NVU and BBB in AD Brains

### 4.1. Bidirectional Pathological Association between Tau and the NVU and BBB

Tau is a microtubule-associated protein that stabilizes microtubules (Figure 2a), polymers of tubulin that form part of the cytoskeleton and provide structure to eukaryotic cells [46]. Tau protein has also been associated with cell signaling, regulation of genomic stability, and synaptic plasticity [47]. Among numerous post-translational modifications, tau is subject to phosphorylation and truncation in AD brain tissue (Figure 2b). Both abnormal changes have been implicated in the accumulation of abnormal tau polymers and paired helical filaments (PHFs) [48,49]. Pathological tau protein leads to microtubule disintegration and NFT formation (Figure 2b) [50,51]. Evidence suggests that dysfunction of the NVU and BBB is triggered by Aβ vascular deposits [52]. However, it has been noted that the presence of pathological tau has been observed as puncta in perivascular spaces in sporadic AD brains [53]. Therefore, dysfunction of the NVU and BBB could trigger tau hyperphosphorylation and, vice versa, tau pathology could induce the NVU/BBB alteration [54] (Figure 2b).

In a transgenic mouse that overexpresses human tau protein carrying the P301L mutation, tau accumulates and eventually leads to BBB disruption [53]. However, the mechanisms by which modified tau could trigger the BBB breakdown are not yet understood. It has been demonstrated that human truncated tau upregulated mRNA expression for several MAPKs (JNK1, p38b, ERK1) and transcription factors (c-Jun, c-Fos, NFkB1, NFkB2). Transcription of pro-inflammatory genes, ultimately leading to the release of proinflammatory cytokines is also increased (Figure 2b). The neuroinflammatory environment has been closely related to BBB dysfunction by increasing permeability, promoting structural changes in brain capillaries, and enhancing the migration of immune cells (Figure 2b) [55]. It has also been suggested that tau-induced activation of glial cells raises the expression of endothelial adhesion molecules and the transport of leukocytes across the BBB, perpetuating the neuroinflammatory environment and exacerbating AD pathology (Figure 2b) [2].

On the other hand, dysfunction of the NVU and BBB could lead to tau pathology. Accumulation of extracellular Aβ induces astrocytes and microglia activation, and the subsequent release of proinflammatory molecules [56]. This neuroinflammatory environment could promote BBB damage and accelerate the development of tau phosphorylation and NFT formation (Figure 2b) [57]. BBB dysfunction can induce pathological tau, and abnormal tau can produce BBB alteration, thus causing a feedback loop (Figure 2b).

### 4.2. CAA Acts as a Trigger for Dysfunction of the NVU

Dysfunction of the NVU and BBB is closely associated with CAA and may exacerbate AD [58]. CAA is distinguished by the accumulation of Aβ fibrils in the walls of capillaries and small to medium-sized arterial blood vessels caliber of CNS parenchyma and leptomeninges. Besides, CAA is strongly associated with spontaneous intracerebral hemorrhage in older adults and a significant risk factor for age-related cognitive decline [59]. CAA has two presentations: sporadic and hereditary. The sporadic CAA is associated with aging and is a common feature of AD, the amyloid is predominantly composed of the Aβ protein [60]. For hereditary CAA, the nature of the amyloid deposits is associated with the respective underlying mutation in the amyloid precursor protein (*APP)* gene [60].

Mutations in the *APP* gene, generally causing single amino acid substitutions, are the triggers of autosomal dominant CAA and early-onset AD. The E693Q (Dutch), L705V (Piedmont), and E693K (Italian) mutations are characterized by severe amyloid angiopathy without NFTs or NPs [61]. Hereditary cerebral hemorrhage with amyloidosis-Dutch type (HCHWA-D), is caused by the E693Q mutation, which causes Aβ aggregation in cerebral and cerebellar meningeal arteries and cerebrocortical arterioles [62]. The histopathological features of HCHWA-D are loss of SMCs, wall thickening, perivascular reactive astrocytes, activated microglia, degenerating neurites, and gemistocytic astrocytes linked to BBB dysfunction [59,63,64,65,66]. These histopathological changes and cognitive deterioration [59] reflect severe vasculopathic changes and severe changes in cellular and structural components of the NVU.

### 4.3. Dysfunction of Components of the NVU in AD

#### 4.3.1. Perivascular Microglial Activation

Microglia are resident innate immune cells of the CNS [67]. Under physiological conditions, microglia cells are in a “resting or quiescent state” characterized by small-bodied cells with long and thin branches. On the contrary, in pathological conditions, they acquire an “amoeboid or activated state,” they swell and retract their processes and are associated with phagocytic activity and pro-inflammatory cytokine release (Figure 3, step 1) [68]. Furthermore, it has been proposed that microglia are the first cells to degrade both soluble and fibrillar Aβ aggregates (Figure 3, step 1) via receptor interaction (Table 1). This interaction could lead to the activation and production of several toxic molecules [69], and finally, alter the BBB/NVU (Figure 3).

#### 4.3.2. Astrocytic End-Foot Dysfunction

The astrocytic end-foot contributes to arteriolar tone by regulating a constant prostaglandin-E2 (PGE2) flow (Figure 3, step 2a) because of the intracellular Ca^2+^ fluctuations [95]. Several astrocytic receptors can interact with the Aβ peptide (Table 1), triggering a neuroinflammatory environment and an alteration in the NVU/BBB [96] (Figure 3, step 2a). During the progression of AD, astrocytic end-foot dysfunction could exacerbate the extracellular and vascular accumulation of Aβ by altering its clearance along with the ISF drainage through the glymphatic system [97]. It favors the CSF flow along with the arterial perivascular spaces, and from there to the brain interstitium through the AQP4. Finally, it removes solutes from the neuropil into the cervical and meningeal lymphatic drainage vessels [98]. Dysfunction of the glymphatic system has been observed in AD animal models, most likely because of the dysfunction of AQP4, which increases both Aβ plaque formation and cognitive deficits [24,98]. In humans, genetic variation in AQP4 affects Aβ burden [99], and AD patients with specific AQP4 single nucleotide polymorphism exhibit a rapidly progressive cognitive decline [100]. Histopathological AD brain studies have demonstrated that the loss of perivascular AQP4 localization predicts disease status [101]. Therefore, a dysfunction of the lymphatic system can cause an impairment of Aβ efflux and its accumulation on brain vessels and dysfunction of the NVU (Figure 3, step 2a).

#### 4.3.3. Pericyte Degeneration

Pericytes are essential for regulating the physical and functional BBB properties through proper signaling with brain endothelial cells (BECs). Both platelet-derived growth factor subunit B (PDGF-B) and platelet-derived growth factor receptor-β (PDGFRβ) are closely involved in signaling in these pericytes/BECs [4,102]. Defective signaling leads to BBB disruption, a decrease in CBF, and hypoxia, which in turn triggers neuronal and synaptic dysfunction (Figure 3, step 2b) [102]. These events may further exacerbate both parenchymal and vascular Aβ accumulation [12]. The interaction between Aβ and the pericytes receptors (Table 1) triggers endothelin-1 release from pericytes, resulting in pericyte contraction and subsequently capillary constriction (Figure 3, step 2b) [103]. There is an association between the accumulation of blood-derived proteins (thrombin, fibrinogen, plasminogen, immunoglobulin G, and albumin) and the pericyte degeneration (Figure 3, step 2b) [104]. Thus, pericyte degeneration can cause a decrease in CBF, leading to neurodegeneration.

#### 4.3.4. Endothelial Cell Degeneration and BBB Breakdown

Postmortem studies in AD brains have shown several vascular alterations, including fragmented vessels, string vessels, irregularities in the capillary surface, changes in vessel diameter, thickening, vacuolization, and local rupture of the capillary basement membrane, associated with vascular Aβ accumulation [46]. Also, data from postmortem human brains have demonstrated a substantial reduction in volume associated with CAA [96]. This event might be partly explained because TJs proteins are substrates of vascular-associated matrix metalloproteinase (MMP) activity (Figure 3, step 2c) [105]. Likewise, low levels of mesenchyme homeobox gene 2 (MEOX2) (Figure 3, step 2c) have been reported in AD brains. MEOX2 has been shown to mediate aberrant angiogenic responses, lead to premature capillary pruning [106], and trigger cerebral hypoperfusion and vascular inflammation [107].

#### 4.3.5. Neuronal Cell Death Mechanisms

In AD, cerebral perfusion is impaired because of the compromised ISF drainage due to the destruction of perivascular space and accumulation of Aβ. Aβ can bind to specific neuronal receptors (Table 1) and cause toxicity, induced mainly by oxidative stress. Oligemia can trigger neuronal dysfunction, altering proteins required for synaptic plasticity [108], and favoring anaerobic brain metabolism. Oligemia can also decrease the generation of ATP, necessary for maintaining the Na^+^, K^+^-ATPase pump and the required action potentials for normal neuronal excitability [109]. It has been shown that transient depletion of oxygen and glucose loss can cause neuronal excitotoxicity and subsequent neuronal death [58]. Oxygen reductions can alter pH and water-electrolyte balance, leading to edema, white matter lesions, and accumulation of both glutamate and Aβ in the brain [29]. A decrease in the levels of GLUT1 (glucose transporter in neurons) and GLUT3 (glucose transporter in BBB) in AD patients suggests that these may contribute to impaired glucose uptake and metabolism in the brain, which could trigger the phosphorylation of tau protein [46] (Figure 3, step 4).

### 4.4. The Two-Hit Vascular Hypothesis and Dysfunction of the NVU

According to the two-hit vascular hypothesis of AD, there exists a vascular contribution (hit one) that followed the effect of Aβ accumulation (hit two) [29]. Vascular risk factors, such as diabetes or hypertension, could lead to NVU/BBB dysregulation and, subsequently, CBF reduction (oligemia), initiating a cascade of events that precedes dementia. This defective Aβ clearance has been postulated to trigger increased cerebrovascular Aβ production and aggregation [110]. There are several receptors in the endothelial cells that promote Aβ clearance (Table 2). The primary receptor that regulates Aβ transcytosis is LRP1 (Figure 3, step 2c; Table 2) [111]. In AD, there are significant reductions in LRP1 [112] and SMCs [113] that correlate with an increase in cerebrovascular Aβ [112]. High levels of serum response factor (SRF) and myocardin in SMCs [113] lead to elevated expression of the sterol response element-binding protein 2 (SREBP2), an important transcriptional LRP1 suppressor, triggering LRP1 depletion and reducing Aβ clearance through the BBB (Figure 3, step 2b) [114]. In contrast, RAGE is the receptor for the influx of Aβ through the BBB that transports it from the blood to the brain (Figure 3, step 2c; Table 2) [115], and an increased expression of RAGE in the brain endothelium of AD patients has been reported (Figure 3, step 2c) [104]. Subsequently, oligemia promotes the CBF loss (Figure 3, step3), resulting in endothelial cell death [116]. Finally, the increase in Aβ accumulation (hit two) accelerates neuronal dysfunction, NFT formation, neurodegeneration, and dementia [29] (Figure 3, step 4).

## 5. Current Status and Challenges for the Pharmacological Treatment of AD

### 5.1. General Limitations in AD Therapy

One of the main hurdles for developing new drugs for AD is that the BBB limits the access of substances to the brain. The endothelium, one of the main components of the BBB, acts as a physical barrier because of the TJs and AJs. Also, it possesses an enzymatic complex (CYP450, xanthine oxidase, MAO) capable of metabolizing various molecules, reducing their bioactivity in the brain. Likewise, efflux pumps (P-gp, MDR and MRP) placed on the endothelial abluminal surface can release molecules back into the peripheral circulation. In AD, the overexpression of the efflux pump may contribute to the failure of therapeutic trials [150]. It is estimated that around 98% of small and 100% of large molecule drugs, do not cross the BBB and hence do not achieve adequate therapeutic concentrations in the brain [151]. Most drugs are incapable of crossing the BBB due to polarity, size, or lack of endogenous transport pathways [152]. Drug delivery systems (DDS) have emerged as an essential procedure to address these concerns. DDS can preserve molecules from degradation [153] and carry them across membranes by hiding their chemical properties but without modifying them [154]. Nanoparticles, as an example of DDS, serve to release drugs for prolonged periods. Nanoparticle transport by receptor-mediated endocytosis is the main procedure to transport drugs across the BBB [155] and this therapeutic approach will be discussed in a section below.

AD therapeutics has focused on replacing the depletion of acetylcholine levels to improve symptoms in patients. Donepezil, one of the most widely used, is an acetylcholinesterase inhibitor that crosses the BBB via the organic cation transporter [156]. Donepezil has been shown to improve cognitive functions in patients with mild to moderate AD by enhancing the acetylcholine bioavailability [157]. However, a major limitation for donepezil is its P-glycoprotein (P-gp)-mediated efflux [158]. There is a threshold for its use: most patients only tolerate low doses because of its adverse cholinergic side-effects. Besides, patients at a particular phase of the disease do not respond to therapy because there is already neuronal deficit or death and hence absence of synaptic connections [159].

### 5.2. Therapeutics Associated with the Amyloidogenic Pathway

Aβ aggregation and binding of Aβ to different receptors appears to be a characteristic hallmark of AD. Therapeutic strategies to decrease Aβ production, promote its elimination, and prevent fibrillar aggregation are being developed worldwide. These have targeted the three enzymes associated with the amyloidogenic pathways by increasing the activity of α-secretase or suppressing β- and γ-secretase (Table 2a–c) [160]. Nevertheless, therapies that modulate β- and γ-secretases have been reported to increase the risk of serious adverse events in AD patients [118,161].

There are two types of immunotherapy directed against Aβ peptides: active (Table 2j) and passive immunization (Table 2l) [162]. While both types of immunotherapeutic approaches against Aβ peptides have shown promise in animal models, these are yet to be translated into the clinic [163]. Typically, active immunotherapy consists of an antigen, alone or conjugated to a non-self T helper cell epitope. Active Aβ immunization produces anti-Aβ antibodies that bind to human NPs and can induce long-term antibody [163]. It can also trigger an immune response by activation of Th2 lymphocytes [164]. Passive immunotherapy involves the direct injection of antibodies and does not require the immune system to generate its own antibody response. This therapy has the advantage that it can be directed at a specific epitope and can be stopped if there are any adverse effects [131]. Passive immunization can also promote Aβ clearance by either central or peripheral mechanisms [164]. In the central mechanism, the anti-Aβ antibodies cross the BBB and bind to NPs, triggering an immune response by activating the Th2 lymphocytes. This culminates in the activation of microglial cells responsible for phagocytosis of Aβ and the reduction of amyloid deposits. In the periphery, antibodies can bind and sequester plasma Aβ. When peripheral Aβ is diminished, Aβ efflux from the brain to the bloodstream is favored, reducing the levels of Aβ in the brain [164]. Nevertheless, passive immunization generally requires expensive humanized monoclonal antibodies and repeated injections, making it less suitable for the long-term treatment of AD than active immunization. Repeated injections of antibodies can also lead to the generation of autoantibodies that may neutralize their effect or even trigger side effects such as vasculitis and glomerulonephritis [163].

Circulating antibodies, following immunization, must be able to cross the BBB to exert their cerebral effect. Few studies have analyzed the mechanisms and ability of antibodies directed against Aβ to enter the brain from the bloodstream. These studies have used IgG antibodies, that have demonstrated low permeability and brain accumulation [165,166,167,168]. It has been suggested that these antibodies can cross the BBB through extracellular pathways [169], since most IgG antibodies have an unsaturated blood transport system to the brain, depending on these pathways to enter CNS [169]. Furthermore, it has been proposed that a brain-to-blood efflux system could exist in the BBB for IgG antibodies [170].

### 5.3. Drugs Targeting Tau Protein

Because pathological tau is both toxic to cells and a mediator of Aβ toxicity [151], targeting tau appears to be a logical therapeutic strategy for AD (Table 2f–i). Several tau aggregation inhibitors have been studied (Table 2f). Methylthioninium, the first tau aggregation inhibitor (TAI) to be discovered [171], was shown to reduce tau pathology and improve cognition in transgenic mouse models of AD and frontotemporal dementia (FTD) [172]. Curcumin, a natural product of the plant Curcuma longa and another TAI, has shown antioxidant and anti-inflammatory properties. However, no significant beneficial effects have been reported in AD subjects receiving curcumin [173].

Tau hyperphosphorylation is observed in AD, and the degree of phosphorylation is reflected by the increased activity of protein kinases [173]. Some kinase inhibitors (Table 2g) prevent cell death and tau phosphorylation, reduce Aβ plaque burden and memory deficits in several models. These compounds can cross the BBB. However, in clinical phases, no clinical improvements have been observed [160,173].

The microtubule stabilizer drugs (Table 2h) have been shown to increase the numbers of microtubules and reduce tau pathology and the number of axons with abnormal morphology in tau transgenic mice. However, some of these drugs have been associated with adverse effects [173].

An antibody targeting the pS396 epitope reduced pathological tau levels and affected the behavioral phenotypes in mice [174,175]. A monoclonal antibody against the epitope in the tau repeat domain inhibited the trans-neuronal propagation of tau aggregation [176]. Several other studies have looked at active and passive immunotherapy approaches in tau mouse models (Table 2i,k) [173]. AADvac1 is an active vaccine consisting of a protein carrier conjugated with a synthetic peptide corresponding to tau [177]. The immunogenicity of AADvac1 in a phase 1 trial has led to a phase 2 trial underway to assess its efficacy in preserving cognitive function [173] (Table 2i). Passive immunization in mouse models has been used to target both extracellular tau and tau aggregates within neurons [173]. Further clinical trials will help establish whether this therapeutic approach is effective in AD (Table 2k).

### 5.4. Non-Steroidal Anti-Inflammatory Drugs (NSAIDs)

Epidemiologic studies have reported that long-term use of NSAIDs reduced the risk for developing AD [178]. The fenamate NSAIDs inhibit the NLRP3 inflammasome activation and decrease microglial activation [179]. Nevertheless, randomized controlled trials investigating the use of NSAIDs and the risk of developing AD did not reach statistical significance among AD patients and subjects without dementia (Table 2n) [180]. Therefore, the effectiveness of NSAIDs for AD treatment remains uncertain.

## 6. Therapeutic Strategies Focused on the NVU

Knowledge of the pathophysiology of the NVU in AD is essential for the implementation of new treatments. Emerging therapies are focusing on promoting or maintaining neural stability by either preventing or restoring damage to the NVU.

### 6.1. Vasculoprotective Effects of Anti-Diabetic, Lipid-Lowering, and Anti-Hypertensive Drugs

According to the two-hit vascular hypothesis of AD, vascular risk factors such as dyslipidemia, obesity, hypertension, and diabetes [181] can trigger dysregulation of the NVU/BBB, leading to dementia [29].

Insulin-resistance has been described in the brain of AD patients [182]. Both insulin and insulin-like growth factor 1 (IGF-1) have multiple brain functions including energy metabolism, trophic support, and synaptic activity [183]. Accumulating evidence suggests that insulin has beneficial effects on the NVU because it is a vasoactive modulator that regulates CBF [184]. Decreased insulin has been associated with endothelial dysfunction, generation of ROS, and excessive free fatty acids released from the adipose tissue [185,186]. Intranasal administration of insulin to the brain has proved feasible in trials with AD patients (Table 3b) [187].

Glucagon-like peptide 1 (GLP-1) analogs are therapeutic agents associated with insulin release and sensitization (Table 3b). The peroxisome proliferator-activated receptors (PPARs) agonists that regulate glucose and lipid metabolism have vasculoprotective and neuroprotective effects, such as regulation of oxidative stress and increasing the CBF and glucose uptake (Table 3b) [204]. Although PPARs were able to improve memory [205], no effect on Aβ pathology was observed in some AD models [204].

Statins (Table 3b) or inhibitors of 3-hydroxy-3-methylglutaryl-coenzyme A (HMG-CoA) reductase are a group of medicines used to treat dyslipidemias, that have been shown to have a neuroprotective effect in an animal model [206]. They can restore vascular reactivity and neurovascular coupling, and improve memory, but do not reduce amyloid pathology [207].

Some antihypertensive drugs can improve NVU function (Table 3a) [208]. For example, third-generation β-blockers can improve endothelial function. Calcium channel blockers (CCBs) can have antioxidant and neuroprotective effects, improve endothelial function and vascular inflammation, and attenuate the neuronal deterioration induced by Aβ [208,209]. The angiotensin receptor blocker (ARBs) and angiotensin-converting enzyme (ACE) inhibitors can improve endothelial dysfunction and vascular inflammation [208]. Other antihypertensive drugs include selective angiotensin II type 1 receptor (AT1) blockers. In rodents, AT1 receptor blockers have potential effects on NVU, such as protecting CBF during a stroke and decreasing neuroinflammation and Aβ neurotoxicity [210]. However, they do not affect soluble Aβ or plaque load in the brain [189].

### 6.2. Drugs for Maintenance of BBB and NVU Integrity

Memantine, an antagonist of extrasynaptic N-methyl-d-aspartate receptors (NMDARs), is currently approved for treatment of AD alone or with acetylcholinesterase inhibitors. NMDARs are ionotropic glutamate receptors that are essential for learning and memory. An increase in NMDAR activation has been related to neuronal dysfunction [211]. Besides, in a mouse model of permanent focal cerebral ischemia, this drug has beneficial effects on NVU because it prevents cerebral ischemia caused by glutamate excitotoxicity, decreases glial activation, and decreases MMP-9 secretion [212]. Therefore, it is considered a drug that in addition to improving cognitive functions can also protect the NVU in AD.

Previous studies have shown that the transforming growth factor-beta (TGF-β), basic fibroblast growth factor (b-FGF), and glial cell-derived neurotrophic factor (GDNF), can promote the BBB integrity (Table 3c) [213].

Proteoglycan NG2, which is a contact adhesion protein that preserves pericyte-brain endothelium interconnections, has been suggested as a possible therapeutic target to maintain correct capillary function [214], and possibly BBB maintenance. Also, drugs that inhibit the metalloproteins effects could keep the BBB integrity [215], limiting the NVU damage.

Notably, a growing body of evidence suggests that interleukin-1β (IL-1β) contributes to BBB dysfunction [216]. Anakinra, an IL-1β receptor antagonist, has been used successfully to treat rheumatoid arthritis [217] and could be a therapeutic target to maintain the BBB integrity in AD (Table 3c). On the other hand, glucocorticoids such as dexamethasone could promote the BBB integrity, triggering the signaling mediated by the glucocorticoid receptor, and regulating the TJs gene expression [218]. Future studies are required to elucidate the beneficial effects of AD.

Multiple sclerosis treatment based on monoclonal antibodies against the α4-integrin receptor on leukocytes [219] (Table 3c) has emerged as a promising therapeutic candidate for avoiding the peripheral immune cell migration into the CNS, reducing neuroinflammation and BBB breakdown in AD.

Targeted inhibition of RAGE–Aβ interaction has been shown to reduce Aβ influx across the BBB, consecutively decreasing the oxidant stress and neuroinflammation environment [82]. However, additional studies with these drugs reporting on long-term outcomes in AD models are required.

### 6.3. Recent Approaches for Improving the NVU

Molecular Trojan horses (MTJs), nanotechnology, and liposomes have emerged as nano-based DDS. DDS offers several benefits for potential drug molecules. These include the ability to preserve molecules from degradation, transport them through the BBB, and conserve their physicochemical properties, and they are biodegradable, biocompatible, nontoxicity, and non-immunogenic [155]. MTJs are non-invasive approaches for drug delivery of large molecules to the brain, using receptor-mediated transcytosis (RMT) [220].

During neuroinflammation, there is an active change with the regulation of pro-and anti-inflammatory signals [221]. Microglia and astrocytes, which constitute the key elements of the NVU, are activated in several neurological diseases, among them AD. Typically, the pro-inflammatory phenotype (M1/A1) corresponds to dysfunction of the NVU [222,223]. It has been suggested that activated microglia induce reactive astrocytes [224]. Reactive astrocytes (A1) can be prompted by cytokines, such as interleukin-1 alpha (IL-1α), TNF-α, and the complement component subunit 1q (C1q), which are secreted by activated microglia (M1) both in vitro and in vivo [224]. The inhibition of these cytokines might result in repair of the NVU and cognitive improvement in AD patients. Therefore, decoy receptors, such as the human tumor necrosis factor receptor (TNFR), have been proposed as novel therapies for neurological disorders [225]. The human TNFR-II extracellular domain as a fusion protein with a chimeric monoclonal antibody (mAb) against the human insulin receptor (HIR) to permit BBB transport of a TNFR decoy insulin receptor has been considered one of the best candidates for RTM-based therapeutics. Human insulin mAb (HImAb) is around 900% more active and ten times more successful than any human transferrin receptor (TfR) [225]. The HIRmAb can function to transport the TNFR therapeutic decoy receptor across the BBB and provide high-affinity binding to human TNFα, suppressing the cytotoxic effects of this cytokine [225]. These properties may be effective in the restoration of the NVU in AD brains.

Nanotechnology is a new field offering an encouraging perspective in the therapy of various CNS disorders, among them AD, and entails the use of materials that have a dimension of between 1 and 100 nm [155]. Nanoparticles cross the BBB, and therefore allow for certain drugs to be delivered to the brain [226]. Human serum albumin (HSA) nanoparticles seem to be a promising therapeutic approach and would allow the administration of some drugs through the BBB [227]. However, future research is required to assess biosecurity and effectiveness for AD.

There is a diversity of natural compounds unable to cross the BBB [155]. They can be administrated through nanoparticles to be effective in AD therapy due to their neuroprotective activity [155]. These beneficial effects on the NVU could result from the inhibition of the Aβ production (by, for example, regulation of secretase activities),increasing Aβ degradation, inhibition of NFT formation, and reduction of neuroinflammation and oxidative stress [228]. The enhancement of a biodegradable nanoparticle was demonstrated for efficient brain accumulation, protection of astrocytes from oxidative stress and mitochondrial alterations using an animal model [229]. This nanomedicine platform has potential beneficial effects on the NVU such as modulating astrocytes to enhance their neuroprotective activities [229].

Several nanoparticles have shown neuroprotective effects through drugs that modulate inflammation and apoptosis, and the neuronal signaling pathways, by down-regulating pro-inflammatory cytokines [230,231] and caspase-3 activity [232], and up-regulating anti-inflammatory cytokines in animal models [230,231]. Their administration resulted in lower mRNA expression of MMP-9, cyclo-oxygenase-2 (COX-2), and inducible nitric oxide synthase (iNOS) [230,231]. All of these molecules are crucial elements for NVU dysfunction [233], and hence promising as targets for AD therapy.

Finally, targeted nanoliposomes and nanoparticles represent a viable and promising strategy for the treatment of AD [234]. They are biocompatible and can carry many therapeutic molecules across the BBB and into NVU cells. Interestingly, immunolabelled liposomes can potentially deliver other novel technologies, such as the mAb MTJs [235]. Modifications of the liposome can include Aβ-targeting ligands (such as curcumin, phosphatidic acid), or retro-inverted peptides, which decrease Aβ and tau aggregation, neuroinflammation, and oxidative stress [234], among others. However, the nonspecific uptake of the cationic liposomes by peripheral tissues and their binding to other proteins limit their therapeutic potential [236]. The intranasal administration of liposomes offers a direct nose-to-brain route. Liposomal encapsulation can facilitate transport across the mucosal barrier. Thus, the intranasal route provides a promising alternative for delivery across the BBB [234].

Additional studies focused on designing new, safe, and effective therapeutic drugs to prevent or restore the NVU damage in AD patients are needed.

## 7. Conclusions and Perspective

Pathological NVU dysfunction has emerged as an essential element in AD pathogenesis. It has been suggested that vascular amyloid deposits can alter the NVU. The dysfunction of the signaling pathways between each of the cells that make up the NVU is associated with a decrease in CBF, hypoxia, neuroinflammatory status, oxidative stress, and finally, degeneration of these cells. It suggests the need to focus on therapies to prevent or repair damage to the NVU. A significant limitation in the treatment for AD is the BBB presented by the endothelial cells. Many drugs cannot cross BBB or are metabolized, decreasing their bioactivity before reaching the brain. Recent therapeutic approaches, that are still at preclinical phases, such as molecular Trojan horses (MTJs), nanotechnology, and liposomes, have been demonstrated to be of use in getting molecules across the BBB and in restoring dysfunction of the NVU. Because the NVU comprises different cellular and structural components, it becomes a real challenge to target them as a therapeutic target. Nowadays, AD therapy has focused on the interaction of limited parts of the NVU. However, it may only be necessary to impact two of the NVU elements to create a domino effect on other components. Further research is required to better demonstrate the mechanisms involved and determine whether immunotherapies or medications aimed at treating cerebrovascular risk factors for AD are able to reduce brain pathology and improve cognitive function. In this review, we have highlighted the physiological and pathological roles of the NVU to identify where avenues for developing innovative treatments for AD may exist.

## Figures and Tables

**Figure 1 ijms-22-02022-f001:**
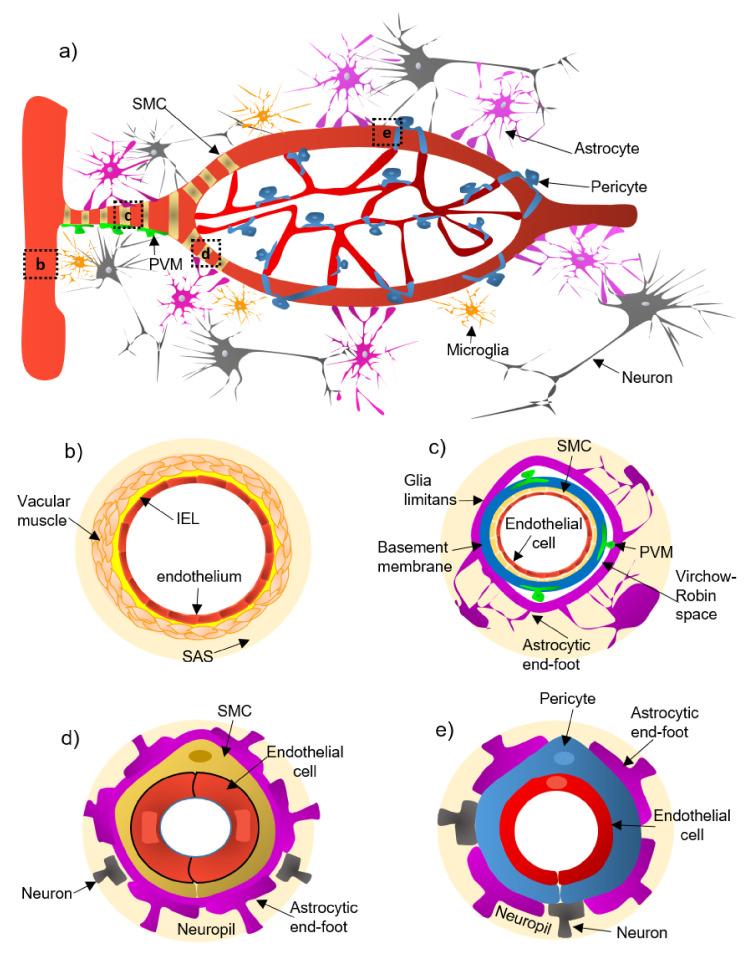
Illustration of the constitutive cellular elements of the neurovascular unit (NVU) along the entire brain vasculature. Panel (**a**) shows a panoramic view of the vascular tree. Panels (**b**–**e**) depict the different levels of the vascular tree. Pial arteries (panel (**b**)) run along the brain surface and penetrate the parenchyma, narrowing and branching into penetrating arterioles (panel (**c**)), which form the intraparenchymal arterioles (panel (**d**)), and ultimately give rise to capillaries (panel (**e**)). Abbreviations: IEL, internal elastic lamina; PVM, perivascular macrophage; SAS, the subarachnoid space; SMC, smooth muscle cell.

**Figure 2 ijms-22-02022-f002:**
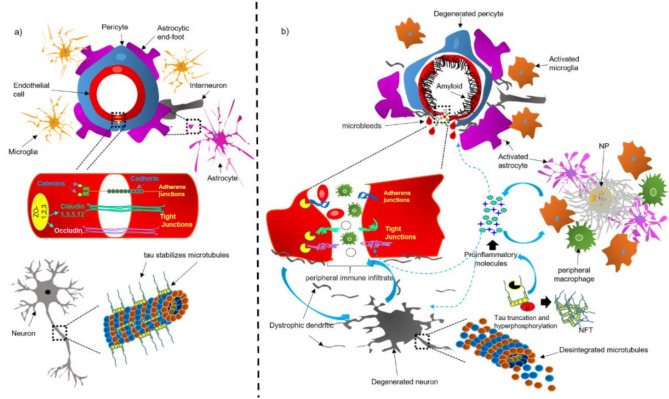
Schematic representation of the NVU under physiological (**a**) and pathological (**b**) conditions. Abbreviations: NFT, neurofibrillary tangle; NP, neuritic plaque.

**Figure 3 ijms-22-02022-f003:**
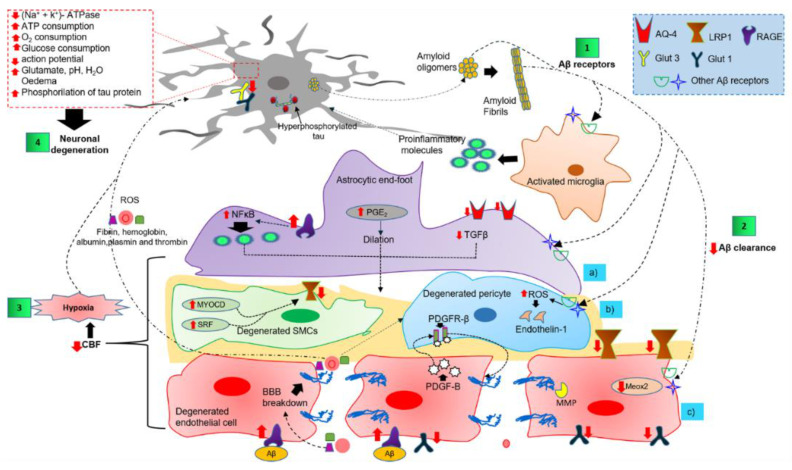
Overview of the complex cell–cell signaling in the NVU, triggered by extracellular and vascular Aβ deposits. Step 1 shows the interaction of Aβ oligomers and fibrils with NVU cells through several receptors. Step 2 illustrates the altered Aβ clearance that triggers NVU dysfunction. Step 3 shows the NVU cell dysfunction as the causative agent of cerebral hypoxia. Step 4 shows cerebral hypoxia, neuroinflammatory environment, and peripheral blood infiltrate as the promoting causes of neuronal degeneration. Abbreviations: AQ-4, aquaporin-4; Aβ, amyloid peptide; BBB, blood-brain barrier; CBF, cerebral blood flow; GLUT, glucose transporter; LRP1, low-density lipoprotein receptor-related protein 1; MEOX2, mesenchyme homeobox 2; MMP, matrix metalloproteinase; MYOCD, myocardin; NFκB, nuclear factor kappa light chain enhancer of activated B cells; PDGF-B, platelet-derived growth factor subunit B; PDGFRβ, platelet-derived growth factor receptor β; PGE_2_, prostaglandin E2; TGFβ, transforming growth factor-beta; RAGE, receptor for advanced glycation end products; ROS, reactive oxygen species.

**Table 1 ijms-22-02022-t001:** Interaction among different species of Aβ aggregates (f: fibrillar; o: oligomeric; m: monomeric; *: unknown), receptors in NVU cells, and pathological effects.

Aβ Receptor and Ligand	NVU Cells	Pathological Effects	Ref.
CR1 (CD35) fAβ	Microglia, astrocytes, neurons	fAβ/CR1 interaction results in C3b/C4b activation. Aβ clearance from the brain via blood cell expresses CR1 in its surface and later metabolism in the liver and/or spleen.	[70,71]
CR3 (Mac-1) fAβ	Microglia	Interaction leads to an increased PI3K/p47PHOX activity (neurotoxicity by superoxide) or NF-kB (inflammatory factors production).	[72,73]
TLR4/6 fAβ	Microglia, astrocytes	CD36/TLR4/6 complex mediates Aβ internalization, followed by ROS and proinflammatory production and phagocytosis.	[74,75]
C5aR (CD88) fAβ, oAβ		C5a/C5aR binding in response to fAβ/oAβ induces TNFα production.	[76,77]
SRA 1/2 fAβ, oAβ		Aβ/SRA interaction results in NF-kB activation and consequently, the secretion of ROS, TNF-a, complement components, among other pro-inflammatory substances.	[78,79]
SRB2 (CD36) fAβ, oAβ	Microglia, BECs, neurons	CD36/a3b1-integrin/CD47 complex regulates fAβ interaction in microglia cells and triggers ROS production, pro-inflammatory cytokines release, and phagocytosis.	[79,80]
RAGE mAβ, fAβ		Aβ/RAGE/p38 and ERK1/2 signaling pathways trigger oxidative stress, NF-kB activation, proinflammatory molecules production, triggering NVU damage.	[81,82]
a7nAChR mAβ, fAβ	Neurons, SMCs, astrocytes	a7nAChR may mediate Aβ internalization. Aβ could activate the JNK/ERK2/MAPK pathway, which results in cell death by apoptosis.	[83,84]
IR mAβ, oAβ	neurons	Aβ/IR binding triggers impaired insulin signaling, which could cause neuronal dysfunction and memory deficits.	[85,86]
SEC-R mAβ	Neurons, glia	Interaction mediates endocytosis and degradation of Aβ by recognizing its 25–35 region.	[87,88]
TREM2 oAβ	Microglia, neurons	Decreased TREM2 leads to Aβ accumulation. TREM2/Aβ linking could trigger neuronal phagocytosis or apoptosis.	[89,90]
LRP1 mAβ	Pericytes, astrocyte, microglia, neurons	LRP1 is widely expressed in NVU cells and mediates the Aβ transport across the BBB. LRP1 controls the Aβ uptake and its subsequent trafficking to the lysosome for degradation.	[91,92]
ABCA1 * Aβ	BECs, pericytes	ABCA1/ApoE/LRP1 complex contributes to brain Aβ transport/clearance. *Abca1* gene deficiency promotes Aβ accumulation in an AD mice model.	[93]
ABCB1 * Aβ		ABCB1/LRP1 transports the Aβ peptides across the BBB. ABCB1 is considered a marker for BBB maturity and functionality.	[94]

Abbreviations: ABCA1, ATP binding cassette A1; ABCB1, ATP-binding cassette sub-family B member 1; ApoE, Apolipoprotein E; BBB, blood-brain barrier; BECs, brain endothelial cells; CR, complement receptor; Cx, complement receptor; fAβ, filamentous Aβ; IR, insulin receptor; LRP1, low-density lipoprotein receptor-related protein 1; NF-kB, nuclear factor-ΚB; NVU, neurovascular unit; oAβ, oligomeric Aβ; p47PHOX, neutrophil cytosol factor 1; PI3K, phosphatidylinositol 3-kinase; ROS, reactive oxygen species; SEC-R, serpin-enzyme complex receptor; SMCs, smooth muscle cells; SRA, Class A scavenger receptor; SRB2, scavenger receptor class B member 2; TLR, Toll-like receptor; TREM2, triggering receptor expressed on myeloid cells 2.

**Table 2 ijms-22-02022-t002:** The effects of potential AD pharmacological therapies on the NVU.

Drug/CT Identifier	Clinical Phase	Effects on the NVU	
**(a) BACE 1 inhibitors**			
E2609/**NCT01294540 ^¥^**	I	Effect on the neuron and BEC. They reduce extracellular Aβ accumulation. Inhibition of the endothelial BACE1 activity may reduce Aβ release into the SMCs and protect vascular wall integrity [117].	[118]
Verubecestat/**NCT01739348 ^¥^**	III		[119]
Umibecestat/**NCT01097096 ^¥^**	III		[120]
LY2886721/**NCT01561430 ^¥^**	III		[121]
**(b) Gamma-secretase inhibitors and modulators**			
Semagacestat/**NCT00594568 ^¥^**	III	Effect on NVU cells. They block Aβ-peptide secretion, precursor agent of NPs, and neuronal dysfunction. According to an animal AD model, some of these drugs can interact with astrocytes and microglia, preventing the Aβ secretion and decreasing the total brain amyloid load [122].	[123]
Avagacestat/**NCT00810147 ^¥^**	II		[124]
Pintol/**NCT00470418 ***	II		[121]
**(c) Alpha-secretase enhancers**			
Epigallocatechin gallate/**NCT00951834, NCT03978052 ^Φ^**	II/III	Effect on NVU cells. Some of these drugs have been shown to improve NVU in animal models, reducing vascular Aβ deposits, neuroinflammation, oxidative stress, synaptotoxicity, and favoring angiogenesis and antioxidant and anti-inflammatory effects [125,126].	[121]
Bryostatin 1/**NCT04538066 ^Φ^**	II		[127]
Etazolate/**NCT00880412 ***	II		[128]
**(d) β-Amyloid aggregation inhibitors**			
Scyllo-inositol (ELND005) **^¥^**	II	Effect on NVU cells. The Aβ polymerization process is a key event involved in AD. Some of these drugs could inhibit Aβ self-association [129]. Therefore, they could inhibit vascular insoluble amyloid deposits and improve the functionality of the NVU in AD.	[130]
Clioquinol (PBT1) **^&^**	III		[121]
Colostrinin *****	III		[121]
**(e) Modulators of beta-amyloid peptide transport-** **RAGE inhibitors**			
Azeliragon (PF-04494700)/**NCT03980730 ^Φ^**	II	Effect on microglia, BEC, and neurons. They regulate the Aβ clearance and improve the BBB functionality, neuroinflammatory environment, and CBF.	[131]
TTP4000/**NCT01548430 ***	I		[132]
**(f) Tau aggregation inhibitors**			
Rember/**NCT00515333 ***	II	Effect on the neuron. Effects on other NVU cells unknown. They prevent oligomeric tau aggregation and NFT formation and disrupt aggregated tau.	[133]
LMTM/**NCT01689246, NCT03539380, NCT03446001 ^Φ^**	III		[133]
**(g) Kinases inhibitors**			
Tideglusib (GSK3β inhibitor)/**NCT00948259, NCT01350362 ***	III	Effect on the neuron. No effect on other NVU cells. Inhibition of kinases leads to tau phosphorylation, loss of affinity for microtubules, NFT formation, and neuronal death.	[133]
Saracatinib (Fyn inhibitor)/**NCT01864655, NCT02167256 ***	II		[133]
Nilotinib (tyrosine kinase inhibitor)/**NCT02947893****^Φ^**	II		[134]
**(h) Microtubule Stabilizers**			
Epothilone D/**NCT01492374 ^¥^**	I	Effect on the neuron. No impact on other NVU cells. They stabilize microtubules and reduce tau pathology and hippocampal neuronal loss.	[133]
Abetotexate/**NCT01966666** *****	I		[133]
**(i) Active immunotherapy anti** **-** **tau protein**			
AADvac-1/**NCT01850238 ***	I	Effect on the neuron. No effect on other NVU cells. It activates T cells and triggers an immune response to eliminate abnormally phosphorylated tau.	[133]
**(j) Active immunity (anti-Aβ peptide polyclonal antibody)**			
Vanutide cridificar/**NCT00479557, NCT01227564 ***	IIa	Effect on the neuron. It induces antibodies against Aβ, preventing Aβ deposition, promoting plaque clearance, and improving cognitive functions.	[135]
**(k) Passive immunotherapy (anti-tau antibody)**			
R07105705 (RG6100)/**NCT02820896, NCT0328914 ^Φ^**	II	Effect on the neuron. No effect on other NVU cells. It recognizes abnormal tau and blocks its transmission from one neuron to another.	[133]
**(l) Passive immunity (anti-Aβ monoclonal antibodies)**			
Aducanumab (IgG1)/**NCT04241068 ^Φ^**	III	Effect on the neuron and possibly on other cells of the NVU. Most of these drugs are directed against epitopes of aggregated forms of Aβ, including soluble monomers, oligomers, and insoluble fibrils. Besides, they favor the central Aβ clearance (except Ponezumab that promotes peripheral release), having a favorable effect on neurons and the other NVU cells. Donanemab, unlike the others, has an affinity for the Aβp3-42 conformation, which is more toxic than Aβ1-40 or 1–42 [136,137]. Furthermore, IgG1 or IgG2 carries a high risk of Fc γ receptor (FcγRs)-mediated overactivation of microglial cells, due to the binding with C1q that can contribute to an inappropriate pro-inflammatory response leading to vasogenic edema and cerebral microbleeds, an effect that does not occur with IgG4. In general, both active and passive immunotherapy tend to have harmful effects on the NVU, increasing the CAA and causing microhemorrhages [138].	[139]
Gantenerumab (IgG1)/**NCT01760005 ^Φ^**	III		[121]
Donanemab (IgG1)/**NCT03367403****^Φ^**	II		[140]
Solanezumab (IgG)/**NCT00905372, NCT01760005 ^Φ^**	III		[141]
Crenezumab (IgG4)/**NCT03977584, NCT01998841 ^Φ^**	II		[142]
Sar228810 (IgG4)/**NCT01485302 ***	I		[143]
GSK933776A (IgG1)/**NCT00459550 ***	I		[144]
Ponezumab (IgG2)/ **NCT00722046,** **NCT00945672** *****	II		[145]
BAN-2401 (IgG1)/**NLT01767311 NCT03887455, NCT04468659 ^Φ^**	IIb		[146]
**(m) Passive immunity (Intravenous immunoglobulin G)**			
Octagam IVIgG/**NCT01300728** **^Φ^**	III	Effects on the NVU cells. They promote central Aβ clearance, block the RAGE receptor, increases sLRP levels, anti-inflammatory effects, and selectively target aggregated Aβ forms (monomers and oligomers).	[147]
Gammagard IVIgG/**NCT00818662 ***	III		[147]
**(n) Nonsteroidal anti-inflammatory drugs (NSAIDs)**			
r-Flurbiprofen **^ζ^**	III	Effects on the NVU cells. γ-secretase inhibitor acts restoring neurogenesis, reorganizing the astrocytic cytoskeleton, reducing pathological tau, rescuing synaptic plasticity, acting on microglia to counteract neuroinflammation	[148]
Itanapraced/**NCT01303744** **^ζ^**	II		[121,149]

**^¥^** Suspended due to adverse effects. * All studies completed but no current active study. **^Φ^** Ongoing clinical study. **^&^** Suspended for toxic environmental effects. **^ζ^** Suspended due to financing/licensing issues. Abbreviations: Aβ, amyloid β-peptide; ACE, angiotensin-converting enzyme; AD, Alzheimer’s disease; APP, amyloid precursor protein; sAPPα, secreted ectodomain APP alpha; ARB, angiotensin receptor blocker; BACE1, beta-site APP cleaving enzyme 1; BBB, blood-brain barrier; BECs, brain endothelial cells; C1q, complement factor; CBF, cerebral blood flow; eNOS, endothelial nitric oxide synthase; Fyn, proto-oncogene tyrosine-protein kinase Fyn; GDNF, glial cell-derived neurotrophic factor; GLP-1, glucagon-like peptide 1; GSK3β, glycogen synthase kinase 3β; HMG-CoA, 3-hydroxy-3-methylglutaryl-CoA; IgG, immunoglobulin G; IL-1β, interleukin-1β; IL-1R1, interleukin 1 receptor type 1; NFTs, neurofibrillary tangles; NPs, neuritic plaques; NVU, neurovascular unit; Pcp, preclinical phase; RAGE, receptor for advanced glycation endproducts; sLRP, soluble low-density lipoprotein receptor-related protein; SMCs, smooth muscle cells; VEGF, vascular endothelial growth factor; γ-PPAR, peroxisome proliferator-activated receptor-γ.

**Table 3 ijms-22-02022-t003:** The effects of potential therapies focused on preventing damage to or restoring function of the NVU in AD.

Drug/CT Identifier	Clinical Phase	Effects on the NVU	Ref
**(a) Antihypertensive drugs**			
Captopril (ACE inhibitor)	Pcp	Effects on the NVU cells. It inhibits Aβ production caused by ACE, regulates pro-inflammatory molecules, and inhibits ROS.	[188]
Losartán (ARBs)/NCT02913664 ^Φ^	II	Effects on the NVU cells. It inhibits ROS production and reduces cerebrovascular/neuropathological changes.	[189]
Amlodipine/NCT02913664 ^Φ^ Nilvadipine/NCT02017340 * (Calcium-channel blockers).	II III	Effects on the NVU cells. They attenuate the neuronal deterioration induced by Aβ, decrease cerebral hypoperfusion due to vasodilator effects.	[190]
**(b) Antidiabetic drugs**			
Exenatide/NCT01255163 ^ζ^ Liraglutide/NCT01843075 ^Φ^ (GLP-1 agonists)	II	Neuron and microglia effects. They promote neuronal survival, synaptogenesis, neurogenesis, anti-inflammation, and protecting against oxidative injury. In AD mouse models reduce Aβ oligomers and plaque load, and microglial activation, improving memory. Also, elicit vasculoprotective effects (protect against hypoxia injury).	[191]
Pioglitazone/NCT01456117 ^†^ NCT00982202 (γ-PPAR) ^†^ Rosiglitazone/NCT00550420 ^†^ NCT00265148 (γ-PPAR) *	III	Effects on the NVU cells. They reduce Aβ levels (enhanced phagocytosis of Aβ deposits), oxidative stress, mitochondrial dysfunction, and neuroinflammation induced by glial cells. They improve cerebral blood flow, and lead to cognitive improvement.	[192]
			[193]
Intranasal insulin/NCT00438568 NCT01547169, NCT01436045 *	II	Neuron effects. They modulate the Aβ levels, protect against synapses damage by Aβ oligomers, and modulate memory consolidation.	[194]
Simvastatine (HMG-CoA)/NCT00053599, NCT00939822, NCT00486044, NCT00303277 *	II	Effects on the NVU cells. They have neuroprotective and pleiotropic impact, improve the vascular system, eNOS activation, and antioxidant effects.	[195,196]
**(c) Novel drugs**			
Natalizumab (antibodies against the α4β1 integrin receptor)	Pcp	Effects on the NVU cells. It acts modulating the peripheral immune system infiltrate into the brain and decrease the proinflammatory environment.	[197]
Anakinra (IL-1 receptor antagonist)	Pcp	Probable effects on microglia cells. It blocks the interaction between IL-1β with its receptor (IL-1R1), then could decrease neuroinflammation in AD patients.	[198]
GDNF/lentiviral vector	Pcp	It has neuroprotector effects against Aβ, protecting neurons and astrocytes.	[199,200]
VEGF	Pcp	Effects on the NVU cells. It modulates angiogenesis, vascular permeability, vascular remodeling, vascular survival, neurotrophic activity, and anti-inflammation.	[201,202]
NGF and NGF mimetics/NCT03069014 *(Intracerebral infusion, nasal/intraocular administration)	II	Effects on the neuron and microglia. They regulate differentiation, growth, survival, and plasticity of cholinergic neurons. They have a direct role in modulating microglial cells toward a non-inflammatory phenotype.	[203]

**^Φ^** Ongoing clinical study. * All studies completed but no current active study. **^ζ^** Suspended due to financing/licensing issues. **^†^** Suspended for lack of effectiveness. Abbreviations: Aβ, amyloid β-peptide; ACE, angiotensin-converting enzyme; AD, Alzheimer’s disease; C.p., clinical phase; eNOS, endothelial nitric oxide synthase; Fyn, proto-oncogene tyrosine-protein kinase Fyn; GDNF, glial cell-derived neurotrophic factor; GLP-1, glucagon-like peptide 1; GSK3β, glycogen synthase kinase 3β; HMG-CoA, 3-hydroxy-3-methylglutaryl-CoA; IgG, immunoglobulin G; IL-1β, interleukin-1β; IL-1R1, interleukin 1 receptor type 1; NFTs, neurofibrillary tangles; NGF, nerve growth factor; NPs, neuritic plaques; NVU, neurovascular unit; Pcp, preclinical phase; ROS, reactive oxygen species; sLRP, soluble low-density lipoprotein receptor-related protein; SMCs, smooth muscle cells; VEGF, vascular endothelial growth factor; γ-PPAR, peroxisome proliferator-activated receptor-γ.

## Data Availability

Not applicable.

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
