# Peer review of "The Neurovascular Unit Dysfunction in Alzheimer’s Disease"

_ijms, 2021, doi:10.3390/ijms22042022_

Round 1

Reviewer 1 Report

This is a concise and detailed review focused on the concept of NVU dysfunction in AD. As the authors suggested, therapeutic approaches based on the NVU dysfunction may shed a light on this currently invincible disease.

1) I suggest some of the subtitles be reorganized. Through the manuscript, the hierarchy of subtitles needs to be rearranged.

E.g. subtitles 4.1 and 4.2 do not match parallelly with each other. And under the subtitle "4.2. CAA causes NVU/BBB dysfunction in AD", five main pathological components of NVU are briefly mentioned. However, all these five main pathologies of NVU dysfunction are not only confined to CAA. CAA may be one main pathophysiology of AD explaining vascular injury to NVU. A more general subtitle may be suitable for this section instead of CAA. Or 1 to 5 sections (1. perivascular microglial ~ 5. Neuronal cell death) may be sufficient as separate sections under "4. NVU and BBB dysfunction in AD brains". And defective Ab clearance may need one separate section.

2) As the authors suggested, all conventional therapeutic approaches focused on the control of vascular risk factors could be considered as "vasculoprotective effect on NVU". However, a more specific mechanism focused on the NVU concept needs to be clarified.

3) Some annotations of the table in the context do not match the table.

4) L353 "active immunization" may be mistyped instead of "passive immunization".

Author Response

Dear reviewer,

Please, find an attached document with the responses to your comments. 

Thank you. 

Reviewer 2 Report

Manuscript ID: ijms-1059097
The Neurovascular Unit dysfunction in Alzheimer's disease.

This is an excellent review of the neurovascular unit dysfunction in Alzheimer's disease (AD). The figures are excellent, summarizing the contents comprehensibly. Since the research points of this area are well-summarized, the manuscript is almost ready for the publication. Below are some suggestions, which may help to improve the value of this review.

<Major Points>
(1) Hereditary Cerebral Hemorrhage with Amyloidosis of the Dutch type (HCHWA-D)
One of the important clinical etiologies relating Abeta to vascular functions may be HCHWA-D, because the mutation in Abeta causes directly the vascular pathologies. The effect of this mutation to the functions of the neurovascular unit may be briefly discussed.

(2) The relationship between the neurovascular unit and the glymphatic system
The drainage (glymphatic) system has attracted the research interests in AD field (for example, Science. 2020 Oct 2;370(6512):50-56). From the bird's eye view, the neurovascular unit and glymphatic system may be an integrated system. A brief comment on the glymphatic system may be valuable.

(3) The permeability of the antibodies against Abeta through BBB
Section 5.2 is an excellent summary of the immunological strategy against AD. However, little is discussed about the permeability of the antibodies against Abeta through BBB. This may be added.

<Minor Points>
(a) line 291, Table 1. Interaction <between> different Aβ aggregation patterns (f: fibrillar; o: oligomeric; m: monomeric; *: unknown), receptors in NVU cells, and pathological effects.
This is an excellent summarizing Table. <between> may be <among>.

(b) line 353, Nevertheless, <active> immunization requires the production of expensive humanized monoclonal antibodies and repeated dosing by injections, making it less feasible for long-term treatment than active 355 immunization.
<active> may be <passive>

(c) Table 2. The effects of potential AD pharmacological therapies on the NVU.
This is an excellent summary of the strategies against AD. However, the information about the effects on NVU are not enough. For example, in b), c), and d) NVU is not referred to.

End of File

Author Response

(The authors gave the same response as above.)

Round 2

Reviewer 1 Report

There still are mistypes, e.g. Table2P in P15, I could not find (p) in table 2. 

Other concerns have been generally well revised according to the comments. 

Author Response

There still are mistypes, e.g. Table2P in P15, I could not find (p) in table 2. 

Other concerns have been generally well revised according to the comments. 

Thank you for your comments.

Sorry for the inconvenience. We had incorrectly referenced a table. The text referred to "Table 3b" instead of “Table 2p”. We have corrected it.

We have revised the text and corrected some errors in English.